# Functionalized Platinum Nanoparticles with Biomedical Applications

**DOI:** 10.3390/ijms23169404

**Published:** 2022-08-20

**Authors:** Sagrario Yadira Gutiérrez de la Rosa, Ramiro Muñiz Diaz, Paola Trinidad Villalobos Gutiérrez, Rita Patakfalvi, Óscar Gutiérrez Coronado

**Affiliations:** Centro Universitario de los Lagos, Universidad de Guadalajara, Lagos de Moreno 47460, Jalisco, Mexico

**Keywords:** platinum nanoparticles, functionalized nanoparticles, nanomedicine, biomedical applications

## Abstract

Functionalized platinum nanoparticles have been of considerable interest in recent research due to their properties and applications, among which they stand out as therapeutic agents. The functionalization of the surfaces of nanoparticles can overcome the limits of medicine by increasing selectivity and thereby reducing the side effects of conventional drugs. With the constant development of nanotechnology in the biomedical field, functionalized platinum nanoparticles have been used to diagnose and treat diseases such as cancer and infections caused by pathogens. This review reports on physical, chemical, and biological methods of obtaining platinum nanoparticles and the advantages and disadvantages of their synthesis. Additionally, applications in the biomedical field that can be utilized once the surfaces of nanoparticles have been functionalized with different bioactive molecules are discussed, among which antibodies, biodegradable polymers, and biomolecules stand out.

## 1. Introduction

Platinum nanoparticles (PtNPs) have gained much attention for their various applications that include the chemical industry, the automotive sector, the biomedical field, and the therapeutic field [1]. In particular, PtNPs have attracted researchers’ interest in their biomedical uses that can be attributed to their excellent biocompatibility, high surface/mass ratio, small size, high reactivity, and electrocatalytic properties [2], which is why they have been used as nanoenzymes due to their similar behavior to superoxide dismutase and catalase [3]. Furthermore, the optical characteristics of PtNPs related to surface plasmon resonance (SPR) make them possible candidates as radiotherapy agents [4]. In relation to other metallic nanoparticles, PtNPs have been reported to have higher catalytic activity compared to palladium nanoparticles [5]; PtNPs have exceptional cellular biocompatibility [6] like gold nanoparticles; however, PtNPs have a greater cytotoxic effect [7], which could be used in cancer therapy since platinum ions can interact with DNA inhibiting its replication [8]; PtNPs have also shown antibacterial activity against Gram-positive and Gram-negative pathogens, which is similar to that of silver nanoparticles (AgNPs), although these present greater toxicity than PtNPs, which limits their clinical use [9].

Nanomedicine is an aspect of nanotechnology that is in constant development due to its various applications in the biomedical field [10,11]. It has been reported that nanoparticles used as nanotherapeutics have a higher desired effect compared to conventional drugs; this can be attributed to surface functionalization, which increases solubility, biocompatibility, and specific targeting capacity [12,13]. Metal nanoparticles can be functionalized by conjugating antibodies, nucleic acids, peptide ligands, and drugs on their surface [14,15]. Recently, researchers have reported the selectivity of functionalized platinum nanoparticles on cancer cells through specific pathways. They can also potentiate radiotherapy as they accumulate specifically in the tumor site. Platinum nanoparticles have also been used in the diagnosis of diseases [16,17,18]. Therefore, this review focuses on recent reports on the functionalization of platinum nanoparticles with bioactive molecules such as antibodies, biocompatible polymers, peptides, and biomolecules, among others, and their biomedical applications.

## 2. Methods for the Synthesis of PtNPs

Some characteristics of PtNPs that are taken into consideration for their industrial and biomedical application are size, morphology, composition, structure, and protection agent [19]. Obtaining biocompatible nanoparticles for biomedical use implies that they are free of contaminants such as organic solvents, endotoxins, Pt precursors, unreacted toxic reagents, etc., which are used in their synthesis [20]. There are various methods that can be used for the synthesis of nanoparticles, which can be classified into physical, chemical, and biological synthesis techniques. We describe each of these in the following sections.

### 2.1. Physical Methods

The physical synthesis of metallic nanoparticles has some advantages: it ensures uniform size distribution, and the nanoparticles have high purity because there is no need to use toxic chemicals. However, these methods require sophisticated instrumentations that require high energy, temperature, pressure, or vacuum, which means they are significantly more expensive than other methods; also, it could be more difficult to adjust the size and shape of nanoparticles [1]. Some examples of the synthesis of PtNPs via physical methods are physical vapor deposition [21,22], the magnetron sputtering method [23], inert gas condensation [24], and pulsed laser ablation in liquid [25]. For example, the PtNPs produced by the laser ablation method (532 nm, 1.6 J/cm^2^ pulse) in which two solvent media were used, sodium dodecyl sulfate (SDS) aqueous solution and pure water, in which the Pt plate was submerged; in aqueous SDS solution, the PtNPs showed a diameter of 2–5 nm and high stability due to their negative charge due to the presence of SDS molecules on the surface; the PtNPs obtained in pure water showed stability without the need to use a surfactant with a diameter of 4–7 nm [26].

### 2.2. Chemical Methods

Chemical syntheses are widely used methods because stable nanoparticle dispersion can be obtained with controlled size and shape; this characteristic of nanoparticles can be achieved with the appropriate selection of solvent, temperature, precursor concentration, and the type and concentration of reducing and stabilizing agents [1]. For example, the most commonly used method for the synthesis of PtNPs is the chemical reduction of platinum cations as precursors with appropriate reducing agents in aqueous or non-aqueous solutions using adequate stabilizing agents. The disadvantages include low purity and the use of toxic chemicals and organic solvents which can be dangerous to humans and the environment. Within chemical synthesis, wet reduction, electrochemical reduction, galvanic displacement, and chemical vapor deposition are among the techniques used. With this method, the physicochemical properties of nanoparticles can be controlled. In particular, wet reduction involves the use of a reducing agent to produce PtNPs from Pt salts in solution, enabling the strict control of morphology and size, achieved by varying the Pt compound concentration, reaction temperature, and the use of polymers, surfactants, and coating agents [1,20,27]. In such fashion, PtNPs were obtained from H_2_PtCl_6_ using ethylene glycol as a solvent, sodium hydroxide as a reducing agent, and polyvinylpyrrolidone (PVP) as a stabilizer; the PtNPs showed different shapes and sizes that depended on the amounts of H_2_PtCl_6_ at a lower concentration. Spherical PtNPs were obtained with a small size (1–2 and 2–3 nm); as the amount of H_2_PtCl_6_ increased, the size increased and the shape of the PtNPs changed to cuboid (5–6 nm), oval (6–8 nm), and flower (16–18 nm) [28].

### 2.3. Biological Methods

Syntheses based on biological processes have been used for the production of monodispersed and stable PtNPs via biosynthesis in bacteria, cyanobacteria, algae, fungi, plant extracts, and bioproducts such as aqueous honey solutions [29]. Biological synthesis or biomolecule-assisted synthesis is commonly used to produce platinum nanoparticles [30]. The advantages of biological methods are that they are cost-effective, environmentally friendly, biocompatible, and easy to apply for large-scale synthesis, and there is no need to use toxic chemicals [1,31]. The disadvantages include the presence of unwanted contaminants such as endotoxins and fragments of biological material, which require difficult, expensive, and time-consuming purification procedures. This ecological method is generally useful for the synthesis of crystalline nanoparticles which vary in shape (spheres, rods, prisms, discs, needles, sheets, or dendrites), while controlling size. These characteristics (shape and size) mainly depend on the parameters of the reaction process, extract, metal salt ratio, pH, temperature, and reaction time [1,20,30,32]. For example, biogenic PtNPs with a spherical shape and a size of 3.47 ± 1.31 nm were obtained using *Nigella sativa* L. extract as reducing agent at a reaction temperature of 75 °C under constant stirring for 48 h [30]. Microorganisms with the ability to reduce Pt (IV) to Pt^0^ have also been used; such is the case of *Acetinobacter calcoaceticus* PUCM 1011 from which PtNPs were synthesized, using a reaction temperature of 30 °C, at a pH of 7, with a concentration of hexachloroplatinic acid (H_2_PtCl_6_) of 1 mM, during 72 h of incubation; these factors influenced the obtaining of cuboidal PtNPs with a size of 2–3.5 nm [33].

## 3. Platinum Nanoparticles Functionalization

To increase the biocompatibility, detection, and specific targeting of nanoparticles, it is necessary to stabilize them to prevent agglomeration and to functionalize them. The most common way to achieve this is by attaching appropriate organic groups to metal surfaces [34]. They can be conjugated with specific units such as low-molecular-weight ligands, peptides, proteins, polysaccharides, polyunsaturated and saturated fatty acids, DNA, plasmids, siRNA, antibodies, tumor markers, and small molecules, increasing their specificity and efficacy (Figure 1) [35,36,37]. Functionalized PtNPs have been used in the biomedical context as anticancer and antibacterial agents in clinical diagnosis and catalysis (Table 1).

### 3.1. Application of PtNPs in Cancer Detection and Treatment

Cancer is one of the leading causes of death worldwide, with more than 10 million new cases being diagnosed each year. Cancer is a multifactorial disease caused by a complex mix of genetic and environmental factors. Recently, a more complete understanding of cancer has been achieved at the genetic, molecular, and cellular levels, providing new goals and strategies for therapy [11]. Cancer treatment options include chemotherapy, radiation therapy, and surgery [38]. Cis-diaminodichloroplatinum (II) (cisplatin) is one of the most widely used chemotherapy drugs for the treatment of multiple types of cancer, including bladder, head and neck, lung, ovarian, and testicular cancer [39]. However, the use of cisplatin is limited due to its toxicity which can cause nephrotoxicity, peripheral neuropathy, and ototoxicity [40]. The use of platinum-based nanomaterials may provide a solution to reduce some side effects of cancer chemotherapies. This is because PtNPs have presented reactivity similar to that of drugs containing Pt [41], since they act as a reservoir of Pt ions that, when entering cells, can induce DNA damage [2]. Its greater surface area improves its catalytic properties, which increases its possible pharmacological use [42]. Recent research has shown that platinum nanoparticles have exhibited anticancer activity against various cancer cell lines, for example, glioma U87 and U251, colorectal HT29, MCF-7 breast, HepG-2 liver, and U937 lymphoma. However, the genotoxic effect of PtNPs was also observed in non-cancerous cell lines [41]. Noble metal nanoparticles (e.g., AgNPs, AuNPs, and PtNPs) can be easily functionalized with antibodies and DNA/RNA to particularly target cells; biocompatible polymers can also be used to prolong their circulation in vivo. Likewise, these nanoparticles can induce cell death through the absorption of light energy or radiofrequencies and convert it into heat, which results in thermal ablation in cancer cells specifically [11]. A combination of treatment with polyphenol tea and cisplatin has been shown to have a synergistic effect by inducing apoptosis by means of caspase-8 and caspase-9 in breast cancer cells, without producing side effects. Due to this, flower-shaped polyphenol-tea-functionalized PtNPs (TPP@Pt) 30 to 60 nm in size were reported [16]. Nanoparticles with this shape are a special category of nanomaterials that have attracted great attention due to the simple preparation procedures, high stability, and improved efficiency due to their large surface/volume ratio that increases their catalytic activity, allowing them to be potential candidates in cancer therapy [43]. Due to these characteristics, the cytotoxic effect of TPP@Pt was demonstrated on cervical cancer cell lines (SiHa); this was dependent on both dose and exposure time. Furthermore, TPP@Pt induces nuclear morphological changes by means of apoptosis [16]. Additionally, these types of flower-shaped nanoparticles have been used in radiotherapy trials on cancer cells. ijms-23-09404-t001_Table 1Table 1Biomedical applications of functionalized PtNP.Nanoparticle TypeFunctionalization MaterialSynthesis UsedNanoparticle SizeApplicationReferenceTPP@PtPolyphenol teaBiological30–60 nmCytotoxic effect in SiHa cells[16] NP-Pt PEGuiladaRhodamine B isothiocyanateChemical34.8 ± 5.3 nmCytotoxic effects in HeLa cells[18] Pt-FAFolic acidChemical10–15 nmCytotoxic effect in HeLa and MFC-7 cells[44] gHPt2.5Peptide gH625Chemical2.5 nmAntioxidant nanoenzyme in HeLa cells[45] Lu-Pt@GSLuminolChemical-Detection of prostate-specific antigen[17] Platinum nanoparticlePVPChemical10–60 nmAntibacterial activity in Gram-positive and Gram-negative strains[32] Ab-PtSelective antibody fragment for penicillinChemical-Colorimetric detection of penicillin in pork[46] 


Metallic nanoparticles have been shown to enhance radiotherapy, beginning with specific accumulation in tumors, thus improving efficacy and reducing toxicity. Biocompatible flower-shaped PtNPs, PEGylated with polyethylene glycol (PGE) diamine and functionalized with rhodamine B isothiocyanate (RBITC) as a fluorescent dye, have been described [18]. PEG is a non-immunogenic biological compound that is made up of repeated units of ethylene glycol. PEGylated nanoparticles guarantee greater biocompatibility and blood circulation time because they prevent phagocytosis by means of the reticuloendothelial system (RES) [47,48]. PEGylated PtNPs functionalized with a fluorescent marker are 34.8 ± 5.3 nm in size. Before carrying out the irradiation study, a colony assay was carried out to determine the cytotoxic effects of the nanoparticles on cervical–uterine cancer cells (HeLa). The fluorescent dye, RBITC, enabled the study of the intracellular localization of the nanoparticles at a concentration of 5 × 10^−4^ mol L^−1^; these were located in the cytoplasm of HeLa cells, where they had not penetrated the nucleus. However, it remains ambiguous whether the RBITC dye was still covalently bound to the PEGylated nanoparticles when they were inside cells. Subsequently, the HeLa cells loaded with the nanoparticles were irradiated with γ rays; the number of cells that survived the presence of the nanoparticles decreased as the radiation dose increased (Figure 2). This decrease was clearly amplified by the presence of nanoparticles, indicating that they enhance the effect of radiation [18].

Another carbon-based nanomaterial that has been used in the functionalization of nanoparticles is graphene oxide (GO) due to the functional groups present on its surface, such as hydroxyl (⁻OH) and carbonyl groups (⁻COOH), which give it solubility in water and polar solvents. In addition, it can be used as a stabilizer and support for nanoparticles thanks to its two-dimensional structure [49,50,51]. Due to these properties, PtNPs have been reported on a GO surface, which acted as a reducing and stabilizing agent. These nanoparticles were also functionalized with folic acid (FA-PtNP/GO). Folic acid can effectively target many tumor cells that overexpress the folate receptor on the cell membrane. Breast cancer cells (MCF-7) and human gastric cancer cells (SGC-7901) were used with human umbilical vein endothelial cells (HUVEC) as a negative control to differentiate the effect of FA-PtNP/GO between cancerous and healthy cells. The results indicated that FA-PtNP/GO selectively bind to target cells through interaction between folic acid and the folate receptor, especially in the membrane of MCF-7 cells, where there is greater overexpression of this receptor [51]. Likewise, MCF-7 cells are useful for the study of breast cancer in vitro, as the cell line preserves several specific characteristics of the mammary epithelium [8]. Additionally, Teow and Valiyaveettil (2010) reported on PtNPs coated with folic acid (Pt-FA) and with PVP (Pt-PVP) to evaluate their toxicity in cancer cell lines. The size of the nucleus of the Pt-PVP particles was in the range of 2–6 nm, whereas those of Pt-FA were found to be larger: 10–15 nm. Pt-FA had more cytotoxic effects on HeLa and MFC-7 cells, which overexpress the folate receptor, than Pt-PVP. Control IMR90 cells that do not overexpress this receptor were affected due to the toxic effect of platinum, even at low doses of Pt-FA. The increase in the toxicity of Pt-FA over Pt-PVP may have been due to the platinum content of 18.2% and 12.5%, respectively, and the increase in endocytosis mediated by the folate receptor. Another factor that may influence the effectiveness of nanoparticles are the Pt^2+^ ions in Pt-FA that can diffuse into the mitochondria and generate reactive oxygen species (ROS), causing cell apoptosis (Figure 3a) [44]. Endocytosis does not ensure the internalization of nanoparticles within cells as it has been reported that they can be retained within endosomes/lysosomes that can degrade them due to low pH and the presence of enzymes, which reduces the therapeutic effect of nanoparticles [45]. The conjugation of bioactive molecules to nanoparticles, such as the peptide gH25, can increase a nanoparticle’s diffusion through the lipid bilayer. This peptide is a derivative of glycoprotein H of herpes simplex virus type 1, and its interaction with the lipid bilayer of cells, which facilitates endocytosis, has been described [52,53]. This was previously demonstrated in an in vitro model described by Guarnieri et al. (2017), in which the ability of the gH625 peptide to facilitate the intracellular administration of PtNPs was tested as a function of particle size (2.5, 5, and 20 nm). To do this, the cellular uptake and intracellular location of the peptide-functionalized PtNPs (gHPt2.5, gHPt5, and gHPt20) and non-functionalized nanoparticles (Pt2.5, Pt5, and Pt20) were evaluated in HeLa cells, and as reported, the uptake of nanoparticles not functionalized with the peptide was dependent on the size, whereas the functionalized nanoparticles showed greater cellular uptake, maintaining greater internalization of the smaller nanoparticles (gHPt2.5). The improvement in cell uptake was due to the fact that the gH625 peptide was capable of fusing with the lipid bilayer through electrostatic and hydrophobic interactions that together were mediators in the penetration of the membrane (Figure 3b). The gHPt2.5 nanoparticles were also observed to be aggregates that entered the cell through clathrin-coated cavities and membrane invaginations (Figure 3c). The percentage of internalization of the functionalized and non-functionalized nanoparticles was calculated: it was 3.3% for gHPt2.5 and 0.5% for Pt2.5. Comparing the results, it can be said that the value of the gHPt2.5 nanoparticles was almost 7 times higher than that of the Pt2.5 nanoparticles. Regarding the functionalized and non-functionalized 20 nm nanoparticles, these were not observed to be free in the cytosol, which indicates that the gH525 peptide had no effect on the transport of nanoparticles larger than (≥5 nm) through the endolysosomal membrane into the cytosol. The antioxidant capacity of the nanoparticles on HeLa cells was also evaluated, and the antioxidant nanoenzymatic activity of gHPt2.5 was confirmed as the production of endogenous reactive oxygen species (ROS) decreased, as well as their overproduction in the face of external aggression induced by hydrogen peroxide [45].

On the other hand, photodynamic therapy (PDT) is used for the production of ROS; these are generated through an excited photosensitizer in the presence of O_2_. The advantages of this therapy are that it has high selectivity on cancer cells and low toxicity compared to conventional treatments [54]. Despite being a non-invasive technique, its efficacy depends on O_2_ levels; however, a hypoxic microenvironment prevails at the tumor site due to tumor proliferation, which would result in unsuccessful PDT and could induce metastasis [55,56]. In addition to PDT, another non-invasive method is photothermal therapy (PTT), which is highly selective with near-infrared laser (NIR)-induced tumor cell ablation [57]. To maximize the efficacy and minimize the side effects, a synergistic therapy combining PDT with PTT has previously been implemented [58]. Therefore, Yang et al. (2021) obtained PtNPs with enzymatic activity similar to catalase with the ability to decompose endogenous H_2_O_2_ into O_2_ and reduce tumor hypoxia, which allowed, by implementing synergistic PDT and PTT therapies, one to produce a greater amount of ROS. Platinum nanoenzymes were fixed on the surface of black phosphorus nanosheets and conjugated with chlorin E6 (Ce6) followed by PEGylation (BP/Pt-Ce6@PEG). The generation of ROS and the efficacy of PTT were evaluated in vitro, using 4T1 murine breast cancer cells, and in vivo, in 4T1 tumor-bearing mice. The functionalized nanoparticles produced a notable amount of ROS, which was evaluated via fluorescence. The amount of ROS was dependent on the exposure time to BP/Pt-Ce6@PEG. Subsequently, the cells were subjected to photoirradiation, so cell viability decreased, reflecting the effectiveness of the nanoparticles. In the in vivo assay, the mice were injected intravenously with BP/Pt-Ce6@PEG. It was confirmed again via fluorescence that the generation of ROS is dependent on the exposure time, and it was suggested that the nanoparticles were deposited in the region of the tumor. Subjecting mice to irradiation inhibited tumor growth, and at the end of the treatment, the tumor was suppressed [56]. PtNPs have also been used to functionalize nanocomposites such as the one reported by Kutwin et al. (2019). In this work, GO platelets prevented the agglomeration of PtNPs and served as an anchor for them as they were distributed between the edges and folds of the GO. The proliferation and viability of the cancer cell lines Colo205, HT-29, HTC-116, SW480 colorectal, HepG2 hepatic, MCF-7 breast, LNCaP prostate, and HeLa B cervical–uterine were evaluated. According to what was reported by the authors, proliferation decreased considerably as a function of the highest concentration, especially in cell lines SW480, HT-29, and Colo205. In addition, cell viability was also affected. Cell deformation was perceived, particularly in Colo205, HepG2, and MCF-7, which could be attributed to the affinity of the nanocomposite to the cell membrane [41]. According to this, it can be said that PtNPs induce an efficient anticancer effect which can be compared with platinum-based compounds [14].

### 3.2. Application of Nanoparticles in the Development of Biosensors

Biosensors are devices that can examine biological samples, and after signal transduction, a quantifiable electrical signal can be obtained. Depending on the signal transducer, they are classified as electrochemical, optical, electronic, piezoelectric, pyroelectric, or gravimetric [59]. The usefulness of biosensors is evident in environmental and bioprocess monitoring, as well as in the food industry with regard to quality control, which allows specific ingredients to be identified and allows contaminants, which can be chemical or biological, to be quantified in agriculture and medicine [60]. Noble metal nanoparticles have shown high catalytic activity and excellent biocompatibility, which is why they are widely used in the development of biosensors [61]. Because of this, PtNPs have also been used for the production of biosensors based on field effect transistors (FETs) for the early detection of breast cancer. This was reported by Rajesh et al. (2016), who developed a biosensor based on graphene oxide field effect transistors (GFETs) as a base for PtNPs that were functionalized with an antibody fragment (scFv) specific to HER3, which is one of four transmembrane receptors that are part of the human epidermal growth factor (HER) receptor family [62]. HER3 overexpression is related to invasive breast carcinomas, and in addition, its detection is linked to resistance to drugs used against cancer, which infers a poor prognosis [63]. The scFv antibody fragments provide an orientation that is superior to classic antibodies, and better tissue penetration has been reported, which is why they are widely used for the recognition of specific tumor markers [64]. Therefore, a modified GFET device with scFv-functionalized nanoparticles was used to quantify the concentration of HER3 in samples with a concentration range from ng/mL to fg/mL. The HER3 detection limit was obtained at 300 fg/mL. These results demonstrate the specificity and effectiveness of the biosensor, which could be implemented in the prompt diagnosis of breast cancer [62]. Additionally, using graphene oxide, Khan et al. (2019) reported a label-free electrochemiluminescent (ECL) immunosensor. This immunosensor was useful for the detection of prostate-specific antigens (PSAs). Levels above 4 ng/mL indicated a high likelihood of developing prostate cancer. The development of the biosensor was carried out using graphene oxide sheets (GSs) modified with 3-aminopropyltriethoxysilane (APTES). The PtNPs were deposited on the GSs and subsequently functionalized with luminol (LuPt@GS) which, in addition to acting as a reducing agent, was also the generator of luminescence. To achieve selectivity, an anti-PSA antibody was conjugated using glutaraldehyde as the cross-linking agent. The PSA was quantified from the reduction in the ECL. To validate the results, first, a calibration curve was made with a standard solution to then detect the concentration of PSA in human serum. As reported by the authors, the ECL immunosensor is highly accurate, selective, and highly sensitive to PSA, so it would be useful in clinical trials [17].

Another application of biosensors focuses on glucose detection. The use of nanomaterials for its elaboration allows the facility to adjust the detection properties; in addition, the use of bimetallic catalysts improves the electrooxidation activity of glucose due to their synergistic relationship [65]. Among the variety of reported biosensors, those using photoelectrochemistry (PEC) can increase target analyte selectivity through photoelectric conversion and an electrochemical transformation. Yang et al. (2022) reported a PEC enzymatic glucose biosensor, in which an electrode based on titanium oxide nanotubes (TiO_2_NTs) was used, in which gold (Au) and platinum (Pt) nanoparticles were deposited for later incorporation of glucose oxidase (GOx). In the presence of glucose, the bimetallic biosensor TiO_2_NTs/Au/Pt/GOx showed a sensitivity of 81.93 µA mM^−1^ cm^−2^, with a detection limit of 1.39 µM. Tests were performed at higher concentrations; however, the stability and selectivity will be reduced at low concentrations (<0.8 mM). So, it can be said that gold and platinum nanoparticles have a synergistic effect on glucose-sensing due to their photocatalytic properties [66].

On the other hand, biomarkers using biofluids have been developed. This is a non-invasive technique for the diagnosis of central nervous system disorders [67]. In view of the above, microRNAs (miRNAs) have been implemented as biomarkers to determine neurodegenerative disorders such as epilepsy, as it has been reported that they may be linked to this disease [68,69]. Epilepsy is a neurological disease characterized by the presence of spontaneous epileptic seizures resulting from synchronous discharges of a neuronal population due to the abnormal dynamism of neural networks [70]. The diagnosis of epilepsy is primarily based on the detailed medical history of the patient, including EEGs and neuroimaging. However, sometimes, this is not sufficient, in which case, biomarkers can contribute to an accurate clinical diagnosis [69]. Because of this, Spain et al. (2015) focused on the design of an electrochemical biosensor for the detection of miRNAs, specifically miR-134, as high levels of this miRNA have been found in patients with epilepsy. For this, PtNPs functionalized with a nucleic acid probe that was specifically for miR-134 were used. Using the electrochemical sensor, the index of miR-134 comprised in the blood plasma of patients with epilepsy and in healthy volunteers was evaluated. Additionally, real-time PCR analysis was performed with an miRNA TaqMan assay. With both methods, the detection of miR-134 with similar values was achieved, even at low concentrations corresponding to the plasma of the group of healthy volunteers; on the contrary, high levels of miR-134 were detected in the plasma of patients with epilepsy. So, the electrochemical sensor based on biofluids can be an alternative method in the diagnosis of diseases [67].

In the food industry, it is important to monitor possible contaminants, which is why laws have been established to protect consumers from toxic products. Due to this, the need arises to analyze and identify new toxic compounds [71]. Such is the case of some antibiotics based on their toxicology and, although they are only indicated for human infections, it cannot be guaranteed that they are not being administered to animals [72]. To identify antibiotic residues in edible animal tissues, the most widely used test is the agar diffusion test; however, characterization can be improved by using immunochemical and electrophoretic methods and microbiological receptor assays [73]. Another method is based on the colorimetric detection of antibiotics of the penicillin family, including penicillin G and V, described by Kwon et al. (2015), which stand out as they are fast and simple. In this immunoassay, clusters of magnetic nanoparticles (MNCs) of Fe_3_O_4_ functionalized with cysteamine (Au/SiO_2_/MNCs or HMNCs) and PtNPs functionalized with a selective antibody fragment for penicillin (Ab-Pt) were used. Antibiotic detection was carried out on a disaggregated pork sample. The function of HMNCs is to chemically capture penicillin antibiotics through the formation of amide bonds between the amine groups of HMNCs and carboxylic acids in antibiotics, allowing Ab-Pt to bind to antibiotic-HMNC complexes. The change in color due to the oxidation of 3,3′5,5′-tetramethylbenzidine (TMB) by Ab-Pt makes it possible to detect antibiotics with the naked eye. Additionally, these interactions can be quantified using light absorption measurements, even at concentrations from 1 ng/mL and over [46]. Using a method similar to that described above, the colorimetric detection of penicillin G was performed on milk samples. This test was carried out in glass vials, which were covered with APTES to produce amine groups at the surface of the vial, while also enabling the capture of antibiotics from the carboxyl group favored by the formation of amide bonds. The analysis was carried out using milk enriched with different concentrations of penicillin G. Subsequently, the platinum dendritic nanoparticles, functionalized with an antibody fragment selective for penicillin G, were added; these were bound to the penicillin captured at the surface of the vial and TMB was added, which induced color changes due to the oxidation of TMB mediated by PtNPs. The color change could be identified with the naked eye with a detection limit of 1 ng/mL [74].

### 3.3. Nanoparticles as Antibacterial Agents

Therapeutic efficacy against infectious diseases depends on traditional treatment based on antibiotics; however, the development of bacterial resistance to multiple drugs limits its effectiveness [75]. Given this scenario, there has been an advance in the production of new drugs that can act as antibacterial agents, which is why it has been shown that metallic nanoparticles are toxic when exposed to pathogenic microorganisms, with platinum nanoparticles among them, which cause severe damage to bacterial cells [1]. Tahir et al. (2017) reported on PtNPs using a green synthesis from the aqueous extract of *Taraxacum laevigatum*, which includes many phenolic biomolecules that acted as reducing and stabilizing agents. According to the characterization results, the surface of the nanoparticles was made up of proteins, flavonoids, saponins, and polyphenols that were found in the extract used in biosynthesis. The antibacterial activity of the nanoparticles against Gram-negative (*Pseudomonas aeruginosa*) and Gram-positive bacteria (*Bacillus subtilis*) was determined. According to the results, the nanoparticles had higher activity against Gram-positive bacteria with an inhibition zone of 18 (±0.8) mm compared to the zone of inhibition of Gram-negative bacteria of 15 (±0.5) mm. The antibacterial capacity of the nanoparticles could be attributed to their size (average 5 nm) and their spherical shape. The antibacterial action of nanoparticles was explained with enzyme denaturation, DNA damage, cell lysis, and the production of hydroxyl (OH^−^) and superoxide (O_2_^−^) radicals (Figure 4) [76].

Additionally, using an aqueous extract, but this time from the brown seaweed *Padina gymnospora*, Ramkumar et al. (2017) reported on PtNPs with a PVP coating, in which the antibacterial activity of platinum nanoparticles with and without PVP coatings was determined against pathogens such as *Escherichia coli*, *Klebsiella pneumoniae*, *Lactococcus lactis*, *Salmonella typhi*, *Staphylococcus aureus*, *Streptococcus mutans*, and *Streptococcus pneuminae*, obtaining greatest activity for the nanoparticles with the PVP coating against *Escherichia coli* with an inhibition zone of 15.6 (±0.17) mm. In the case of the other strains, the inhibition zone was reached between 13 and 14 mm. In contrast, the nanoparticles without the PVP coating did not have the ability to inhibit bacterial growth, showing a zone of inhibition between 2 and 3 mm, which may have been due to the *P. gymnospora* extract [32].

As discussed above, the biological synthesis for obtaining nanoparticles is one of the most used methods because it does not require the presence of chemicals that can be toxic; instead, plant extracts are used as reducing agents and stabilizers [30]. Therefore, Subramaniyan et al. (2018) used phytoproteins extracted from spinach leaves to functionalize PtNPs. Its antibacterial activity against *Salmonella typhi* was evaluated, consequently, due to the reports of food intoxication related to this bacterium. A minimum inhibitory concentration of 12 µM was obtained. In the susceptibility test, an inhibition halo of 13 mm was achieved. In vivo model was carried out using zebrafish, which were exposed to *Salmonella typhi*, and after infection, functionalized nanoparticles were injected; according to the authors, the fish recovered successfully and without sequelae. Based on the obtained results, it can be said that functionalized platinum nanoparticles with phytoproteins can be a promising candidate to combat infectious diseases caused by *Salmonella typhi* [77].

## 4. Conclusions

PtNPs are materials of high importance in the biomedical field due to their multiple applications. The functionalization of these nanomaterials improves them in terms of bioactivity, due to which they stand out as anticancer agents, as they have shown a specific cytotoxic effect in in vitro tests. Their behavior is similar to that of enzymes, making them important components in the manufacturing of useful biosensors for the rapid diagnosis of diseases and the detection of substances of interest with a colorimetric change through oxidation. Likewise, their antioxidant capacity has been demonstrated, as they reduce oxidative stress through the elimination of intracellular ROS. The conjugation of specific molecules with the surface of nanoparticles has also been reported to improve entry into cells by increasing their bioavailability. Similarly, functionalization makes them promising candidates as antibacterial agents due to their inhibitory action against Gram-positive and Gram-negative pathogens. The differences between the methodologies used to obtain platinum nanoparticles were discussed here, as well as the advantages and disadvantages in their synthesis. In general, it was shown that PtNPs have various applications in nanomedicine, and their functionalization is a field that is in constant development, so it is probable that in the future, more reports will be published regarding these nanomaterials.

## Figures and Tables

**Figure 1 ijms-23-09404-f001:**
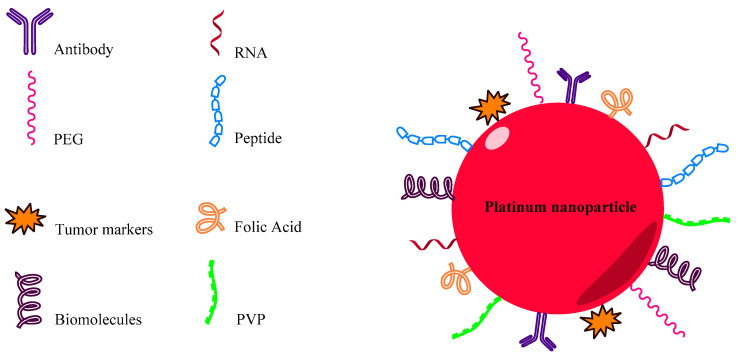
Functionalized PtNP through conjugation of antibodies, peptides, nucleic acids, tumor markers, biomolecules, folic acid, and polymers.

**Figure 2 ijms-23-09404-f002:**
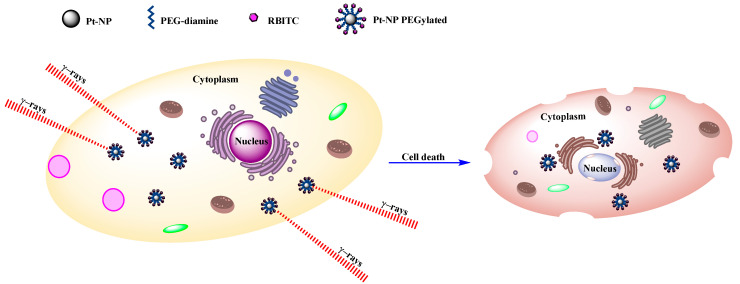
Gamma-ray-induced cell death amplified by PEGylated PtNPs.

**Figure 3 ijms-23-09404-f003:**
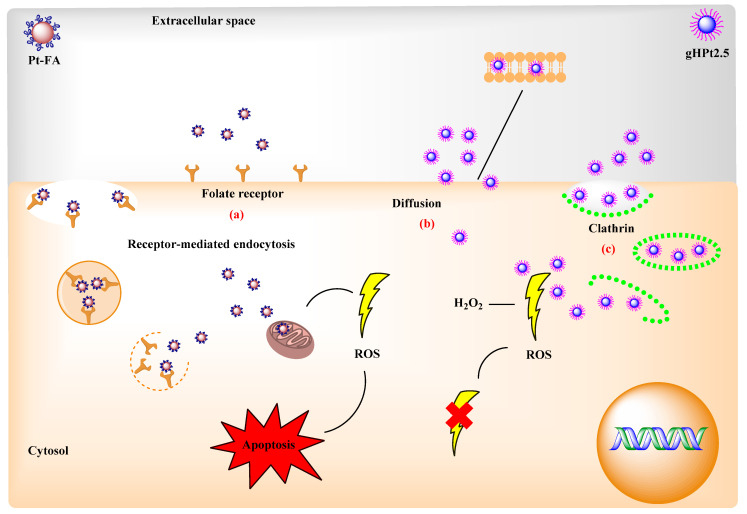
The methods of PtNPs uptake into a cell. (**a**) Folate-receptor-mediated endocytosis. (**b**) Diffusion. (**c**) Clathrin-mediated endocytosis.

**Figure 4 ijms-23-09404-f004:**
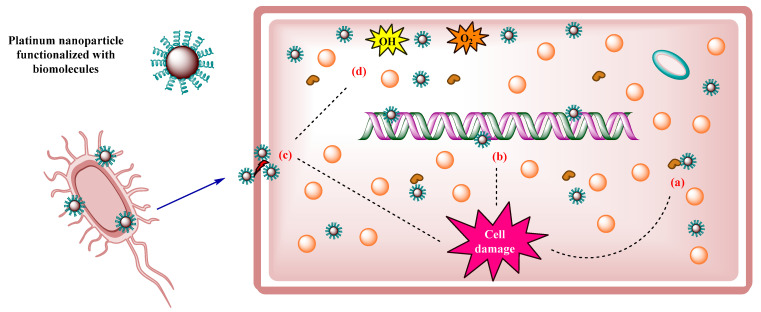
Mechanism of action of PtNPs against bacterial cells. (**a**) Denaturation of essential enzymes. (**b**) DNA damage. (**c**) Cell lysis. (**d**) Production of hydroxyl radicals (OH) and superoxide (O_2_^−^).

## Data Availability

Not applicable.

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
