# Peer review of "Functionalized Platinum Nanoparticles with Biomedical Applications"

_ijms, 2022, doi:10.3390/ijms23169404_

Round 1
Reviewer 1 Report
The manuscript entitled "Functionalized platinum nanoparticle with biomedical applications" provides an overview of research on the synthesis and application of platinum nanoparticles. The article is written in good language, but the review is quite brief. It is proposed to adopt it after minor changes.
1) Line 43 - the title is separated from the text.
2) Line 83 - it is recommended to add a link to the source, for example, https://doi.org/10.3390/mi12121480 .
3) Extra lines - 47, 58, 76, 91, 102, 277, 380.
4) It might be appropriate to slightly increase the scope of the review by adding chapters and references to literary sources, at the discretion of the authors.
Author Response
Point 1: Line 43 - the title is separated from the text.
Response 1: The manuscript was modified
Point 2: Line 83 - it is recommended to add a link to the source, for example, https://doi.org/10.3390/mi12121480 .
Response 2: the recommended bibliography was included, line 105
Point 3: Extra lines - 47, 58, 76, 91, 102, 277, 380.
Response 3: The manuscript was modified
Point 4: It might be appropriate to slightly increase the scope of the review by adding chapters and references to literary sources, at the discretion of the authors.
Response 4: Based on the comments, the bibliography was increased, and the content of some chapters of the manuscript was increased.
Reviewer 2 Report
The subject of the review is very interesting but some discussions are poorly argued, as for example in the paragraph 2 "methods for the syntesys of platinum nanoparticles" where the authors have to described the different methods "Physical, chemical and biological" and not just list them.
In the text "platinum nanoparticles" is reported as Pt nanoparticles in the row 55 but in the text only in the paragraph 2.2 are reported as such, in all the text are indicated as platinum nanoparticles, so it would be better to use Pt the beginning (row 29) and so on in all the text
in the paragraph "chemical methods" is reported many times the same concept "size and shape controlled" (see row 62, 65-66, 70,73)
In table 1 are reported the references 31-33 but the number 30 is reported in the text only after (row 119)
In the Figure 1 is reported as caption "biomaterials" but it is impossible to define Antibody and also others as such
Check row 151 and 152 (TPP and not TTPP)
In the Figure 2 is not so clear by the image the cell death
In the Figure 3 it would be better to indicate "diffusion" instead of "transportation by diffusion"
In the row 254 PTT instead of phototherapeutic therapy
Author Response
Based on your comments and the observations of the other referee, we have modified the manuscript.
Point 1: The subject of the review is very interesting but some discussions are poorly argued, as for example in the paragraph 2 "methods for the syntesys of platinum nanoparticles" where the authors have to described the different methods "Physical, chemical and biological" and not just list them.
Response 1: The manuscript was modified and the lines added line 54-58.
“Some characteristics of PtNPs that are taken into consideration for their industrial and biomedical application are size, morphology, composition, structure, and protection agent [19]. Obtaining biocompatible nanoparticles for biomedical use implies that they are free of contaminants such as organic solvents, endotoxins, Pt precursors, unreacted toxic reagents, etc., which are used in their synthesis [20].”
Point 2: In the text "platinum nanoparticles" is reported as Pt nanoparticles in the row 55 but in the text only in the paragraph 2.2 are reported as such, in all the text are indicated as platinum nanoparticles, so it would be better to use Pt the beginning (row 29) and so on in all the text
Response 2: The manuscript was modified
Point 3: in the paragraph "chemical methods" is reported many times the same concept "size and shape controlled" (see row 62, 65-66, 70,73)
Response 3: The manuscript was modified, the text was rewritten new lines 77 to 89
“Chemical syntheses are widely used methods because stable nanoparticle dispersion can be obtained with controlled size and shape, this characteristic of nanoparticles can be carried out with the appropriate selection of solvent, temperature, precursor concentration, and the type and concentration of reducing and stabilizing agents [1]. For example, the most commonly used method for the synthesis of PtNPs is the chemical reduction of platinum cations as precursors with appropriate reducing agents in aqueous or non-aqueous solutions using adequate stabilizing agents. The disadvantages include low purity and the use of toxic chemicals and organic solvents which can be dangerous to humans and the environment. Within chemical synthesis, wet reduction, electrochemical reduction, galvanic displacement, and chemical vapor deposition are among the techniques used. With this method, the physicochemical properties of nanoparticles can be controlled. In particular, wet reduction involves the use of a reducing agent to produce PtNPs from Pt salts in solution, enabling the strict control of morphology and size, achieved by varying the Pt compound concentration, reaction temperature, and the use of polymers, surfactants, and coating agents [1,20,27].”
Point 4: In table 1 are reported the references 31-33 but the number 30 is reported in the text only after (row 119)
Response 4: The bibliography was numbered correctly in the text and in the table.
Point 5: In the Figure 1 is reported as caption "biomaterials" but it is impossible to define Antibody and also others as such
Response 5: The manuscript was modified
New foot of figure line 130-131:
“Functionalized PtNP through conjugation of antibodies, peptides, nucleic acids, tumor markers, biomolecules, folic acid and polymers”.
Point 6: Check row 151 and 152 (TPP and not TTPP)
Response 6: The manuscript was modified, lines 163-165
“Due to these characteristics, the cytotoxic effect of TPP @ Pt was demonstrated on cervical cancer cell lines (SiHa); this was dependent on both dose and exposure time. Furthermore, TPP @ Pt induce nuclear morphological changes by means of apoptosis [16].”
Point 7: In the Figure 2 is not so clear by the image the cell death
Response 7: The image was modified to appreciate gamma-ray-induced cell death amplified by PEGylated PtNPs.
Point 8: In the Figure 3 it would be better to indicate "diffusion" instead of "transportation by diffusion"
Response 8: The figure was modified
Point 9: In the row 254 PTT instead of phototherapeutic therapy
Response 9: The manuscript was modified, Line 265
Reviewer 3 Report
A manuscript by Sagrario Yadira Gutiérrez de la Rosa reviews the current knowledge of functionalized Platinum nanoparticles' application in medicine. The topic is important and worth publishing. However, In my opinion, the manuscript should be improved before acceptance. The selection of content is generally all right, however, the organization of the manuscript fails, as well as the logic structure is problematic. Below I pointed out some examples I found (but please, refer to them as the examples, I did not list all, as they are repeating).
1) introduction starts with „nanomedicine” while it is not the title or special purpose. It should be reorganized, to start with Platinum nanoparticles and maybe then mention, that they are important in nanomedicine.
2) define platinum nanoparticles as Pt nanoparticles earlier than in line 54. Maybe PtNPs would be a more useful abbreviation?
3) a general overview of Pt nanoparticles properties, in comparison to other NPs seems to be missing. I mean defining basic size ranges, shape, and absorption properties (lack of fluorescence? plasmon effects? etc).
4) subchapter „Biomedical applications of functionalized platinum nanoparticles” start in the way by suggesting, that this is rather the subchapter about functionalization, not an application. Please, rewrite the chapter or adjust the title.
5) There are a lot of general statements, which are correct but need more details. E.g. lines 111-112. The reader does not know what side effects, what types of chemotherapies, etc. Lines 113-115 are not the logical continuation, since the reader was not introduced to the issues of surface/mass ratio and other physicochemical characteristics, as well as the bioavailability.
I’m not sure what the bioavailability here means.
6) line159 - PEG abbreviation should be explained earlier.
7) Table 1 - I don’t understand the heading „Biological activity of nanoparticles” and the e.g. Assays of cell viability, nuclear morphology, and cell cycle distribution in SiHa cells. How an assay may be an activity? or what is actually „nuclear morphology” - is it changed after PtNPs action? The same refers to the whole column in the table.
8) line 142-143 - I guess, the Authors do not mean that biocompatible polymers can convert light. please, rewrite, this is only one example of many similar sentences.
9) line 145 - all metallic nanoparticles?
10) line 149 - resistance to what?
11) line 177 - again, do you refer specifically to Pt nanoparticles or all types of nanoparticles? Please also remember, that graphene oxide is a nanoparticle by itself.
In general, the manuscript would benefit from shorter paragraphs, dealing with one specific example (maybe now it is just the formatting error, but there are continuous sections about many not clearly related examples).
Author Response
Based on your comments and the observations of the other referee, we have modified the manuscript.
Point 1: introduction starts with “nanomedicine” while it is not the title or special purpose. It should be reorganized, to start with Platinum nanoparticles and maybe then mention, that they are important in nanomedicine.
Response 1: The manuscript was modified, new lines: 24-29.
“Platinum nanoparticles (PtNPs) have gained much attention for their various applications that include the chemical industry, the automotive sector, the biomedical field and in the therapeutic field [1]. In particular, PtNPs have attracting researchers’ interest for their biomedical uses that can be attributed to their excellent biocompatibility, high surface/mass ratio, small size, high reactivity, and electrocatalytic properties [2], which is why they have been used as nanoenzymes due to their similar behavior to superoxide dismutase and catalase [3].”
Point 2: define platinum nanoparticles as Pt nanoparticles earlier than in line 54. Maybe PtNPs would be a more useful abbreviation?
Response 2: the abbreviation was modified in the manuscript.
Point 3: a general overview of Pt nanoparticles properties, in comparison to other NPs seems to be missing. I mean defining basic size ranges, shape, and absorption properties (lack of fluorescence? plasmon effects? etc).
Response 3: The manuscript was modified, new lines 30-39.
“Furthermore, the optical characteristics of PtNPs related to surface plasmon resonance (SPR) make them possible candidates as radiotherapy agents [4]. In relation to other metallic nanoparticles, PtNPs have been reported to have a higher catalytic activity compared to palladium nanoparticles [5]; PtNPs have exceptional cellular biocompatibility [6] like gold nanoparticles, however, PtNPs have a greater cytotoxic effect [7], which could be used in cancer therapy since platinum ions can interact with DNA inhibiting its replication [8]; PtNPs have also shown antibacterial activity against gram-positive and gram-negative pathogens, which is similar to that of silver nanoparticles (AgNPs), although these present greater toxicity than PtNPs, which limits their clinical use [9].”
Point 4: subchapter „Biomedical applications of functionalized platinum nanoparticles” start in the way by suggesting, that this is rather the subchapter about functionalization, not an application. Please, rewrite the chapter or adjust the title. (Line 88)
Response 4: The manuscript was modified, new title:
“Platinum nanoparticles Functionalization”, line 119.
Point 5: There are a lot of general statements, which are correct but need more details. E.g. lines 111-112. The reader does not know what side effects, what types of chemotherapies, etc. Lines 113-115 are not the logical continuation, since the reader was not introduced to the issues of surface/mass ratio and other physicochemical characteristics, as well as the bioavailability.
I’m not sure what the bioavailability here means.
Response 5: The manuscript was modified, new lines 137-146.
“Cancer treatment options include chemotherapy, radiation therapy, and surgery [38]. Cis-diaminodichloroplatinum (II) (cisplatin) is one of the most widely used chemotherapy drugs for the treatment of multiple types of cancer, including bladder, head and neck, lung, ovarian, and testicular [39]. However, the use of cisplatin is limited due to its toxicity which can cause nephrotoxicity, peripheral neuropathy, and ototoxicity [40]. The use of platinum-based nanomaterials may provide a solution to reduce some side effects of cancer chemotherapies. This is because PtNPs have presented a reactivity similar to that of drugs containing Pt [41], since they act as a reservoir of Pt ions that, when entering cells, can induce DNA damage [2]. Its greater surface area improves its catalytic properties, which increases its possible pharmacological use [42].
Point 6: line159 - PEG abbreviation should be explained earlier.
Response 6: The manuscript was modified, line 173.
Point 7: Table 1 - I don’t understand the heading „Biological activity of nanoparticles” and the e.g. Assays of cell viability, nuclear morphology, and cell cycle distribution in SiHa cells. How an assay may be an activity? or what is actually „nuclear morphology” - is it changed after PtNPs action? The same refers to the whole column in the table.
Response 7: The table was modified according to the recommendation.
Point 8: line 142-143 - I guess, the Authors do not mean that biocompatible polymers can convert light. please, rewrite, this is only one example of many similar sentences.
Response 8: The manuscript was modified, new lines: 153-155.
“Likewise, these nanoparticles can induce cell death through the absorption of light energy or radiofrequencies and convert it into heat, which results in thermal ablation in cancer cells specifically [11].”
Point 9: line 145 - all metallic nanoparticles?
Response 9: The manuscript was modified, new line: 150-153.
“Noble metal nanoparticles (e.g., AgNPs, AuNPs, and PtNps) can be easily functionalized with antibodies and DNA/RNA to particularly target cells, biocompatible polymers can also be used to prolong their circulation in vivo.”
Point 10: line 149 - resistance to what?
Response 10: New lines 159-162.
“Nanoparticles with this shape are a special category of nanomaterials that have attracted great attention due to the simple preparation procedures, high stability, and improved efficiency due to their large surface/volume ratio that increases their catalytic activity, allowing them to be potential candidates in cancer therapy [43].”
Point 11: line 177 - again, do you refer specifically to Pt nanoparticles or all types of nanoparticles? Please also remember, that graphene oxide is a nanoparticle by itself.
Response 11: The manuscript was modified, new line 191
Round 2
Reviewer 3 Report
I am satisfied with changes introduced by Authors.
I would only suggest modification of figure 3 and 4 titles, to start with a general title - e.g. The ways of nanoparticles uptake into a cell for figure 3. then, of course, current points a-d should follow.
Author Response
Based on your comments and the observations of the other referee, we have modified the manuscript.
Point 1: I would only suggest modification of figure 3 and 4 titles, to start with a general title - e.g. The ways of nanoparticles uptake into a cell for figure 3. then, of course, current points a-d should follow.
Response 1: The manuscript was modified, new lines: 248
“Figure 3. The ways of PtNPs uptake into a cell. a) Folate-receptor-mediated endocytosis. b) Diffusion. c) Clathrin-mediated endocytosis.”
New lines: 418
“Figure 4. Mechanism of action of PtNPs against bacterial cells. a) Denaturation of essential enzymes”.
Dear referee, we greatly appreciate all your comments, we hope that our answers are adequate, so that you consider our article for publication. Best regards.